# The role of international support programmes in global health security capacity building: A scoping review

Anne Doble[1,2], Zoe Sheridan[1]*, Ahmed Razavi[1], Anne Wilson[1], Ebere Okereke[3]

1 International Health Regulations Strengthening Project, Global Operations Directorate, UK Health Security Agency: Nobel House, London, United Kingdom, 2 Health Education England North West: 3 Piccadilly Place, Manchester, United Kingdom, 3 Tony Blair Institute for Global Change: 1 Bartholomew Close, London, United Kingdom

* zoe.sheridan@ukhsa.gov.uk

**Data Availability Statement:** As this is a narrative scoping review, no primary data was collected/ used for the study. Secondary data sources are cited in the References. The search methodology is

## Abstract

Large scale public health emergencies such as COVID-19 demonstrate the importance of Global Health Security (GHS) and highlight the necessity of resilient public health systems capable of preparing for, detecting, managing, and recovering from such emergencies. Many international programmes support low- and middle-income countries (LMICs) to strengthen public health capabilities for compliance with the International Health Regulations (IHR). This narrative review seeks to identify key characteristics and factors necessary for effective and sustainable IHR core capacity development, establishing roles for international support and some principles of good practice. We reflect on the "what" and the "how" of international support approaches, highlighting the importance of equitable partnerships and bi-directional learning, and inviting global introspection and re-framing of what capable and developed public health systems look like.

## Introduction

Acute public health emergencies can have devasting impacts in terms of lives lost, morbidity, and development and economic costs, also severely testing health systems and exacerbating weaknesses. These consequences highlight the importance of global health security (GHS), defined by the World Health Organization (WHO) as "the [proactive and reactive] activities required. . . to minimize the danger and impact of acute public health events that endanger people's health across geographical regions and international boundaries" [1]. The International Health Regulations (2005) (IHR) are the governing framework for GHS, outlining responsibilities for WHO and member states, and defining core competencies for effective prevention, detection, control and response to public health threats [2–4].

Despite evident progress in developing these public health capacities, responses to recent epidemics–including Ebola, Zika, influenza [5], and notably COVID-19 –indicate that IHR implementation and compliance remain a substantial challenge worldwide and particularly in low- and middle-income countries (LMICs) [5, 6]. Joint external evaluations (JEE)–a non-statutory peer-evaluation of IHR core capacities forming part of the WHO IHR Monitoring and

described in the main article. The S1 Appendix also provides full details of the search strategy/terms, and a PRISMA checklist and flowchart for scoping reviews.

**Funding:** The authors received no specific funding for this work.

**Competing interests:** I have read the journal's policy and the authors of this manuscript have the following competing interests: EO currently serves on the editorial board of PLOS Global Public Health. This will not alter adherence to PLOS Global Public Health policies on sharing data and materials. The authors have declared that no other competing interests exist.

Evaluation Framework (IHR MEF)–highlight gaps in key areas in many countries [7–10]. The impacts of these capacity gaps and limited health system resilience have been vividly demonstrated through the COVID-19 pandemic, in both LMICs and high-income countries (HICs) [11–16]. Possible explanations for failures to address these gaps include de-prioritisation and underfunding of IHR capacity-building efforts in the face of competing priorities [9], or a failure to adopt evidence-based approaches informed by local need, within IHR capacity-building initiatives. Other factors hindering these efforts may include failure to adopt "all hazards" perspectives or recognise the multisectoral nature of health emergencies, fragmentation of public health systems, and underinvestment in health system strengthening [10, 17]–as explored subsequently. Thus the need for renewed and sustainable investment to build public health capacity for improved GHS is clear [9, 12, 14].

International bilateral, multilateral and non-government organisation (NGO) funded programmes have long been involved in delivering "technical assistance" [18], capacity building [19] or partnership support in LMICs. Often originating in HICs, such international support programmes can play a key role in assisting LMICs to address IHR-related public health capacity gaps [9, 20]. The WHO Strategic Partnerships for IHR (2005) and Health Security Portal identifies over 60 bilateral or multilateral partners engaged in financial and/or technical support for capacity development across the 19 IHR technical areas, through investments amounting to almost $8 billion (USD) [21]. The UK IHR Strengthening Project is one such bilateral programme, providing technical partnership to LMICs and regional bodies to build capability, for improved GHS and public health system resilience [22].

For sustainable impact, international partnership support efforts and programmes should adopt evidence-based approaches and contribute to the evidence base. Here we review the peer-reviewed and grey literature to ascertain factors associated with effective, equitable, and sustainable capacity building for improved IHR compliance in LMIC settings–specifically focussing on the role of international partners. We discuss learning thematically, identifying the "what" and the "how" of impactful IHR capacity building approaches. We sought not to impose predetermined themes according to an existing framework. Rather the conceptual themes under the "what" and "how" are presented as they emerge, working approximately from a micro or small-scale to a macro or systemic focus. Furthermore, whilst recognising national public health functions and capacities can be extremely diverse, this review focuses on those relating to health security and IHR core capacities specifically [2, 7]. This review has also enabled the IHR Strengthening Project to re-examine its approaches and underlying assumptions–promoting continuous learning, adaptive programming, and evidence-based action.

Such evidence reviews are timely as the COVID-19 pandemic has exposed weaknesses in 'strong' and well-prepared public health systems in LMICs and HICs [14, 15], and as the fitness of the current IHR (2005) is being scrutinised [12, 14, 23]. The pandemic has shown that demonstrating technical IHR capacities alone is insufficient to achieve GHS [4, 24]; assumptions are also being challenged on what health system resilience is and how it can be attained [15, 25]. Evidently improved global collaboration is crucial and HICs (and HIC-led support programmes) have much to learn from LMIC partners [16, 25].

## Search strategy and selection criteria

The literature search strategy for this narrative scoping review was based on the following research questions:

- Which LMICs have strengthened their public health systems, global health security capacity and IHR compliance, and how was that achieved?

- What was the role of aid-funded programmes in achieving that outcome, and what approaches did they apply?

Two alternative search strategies were initially applied, with the more simplified strategy ultimately adopted as this produced a larger pool of results. GHS and IHR capacity building, and low- and middle-income country related search terms (including synonyms) were used in combination to identify relevant LMIC-focussed publications, searching within subject headings, keywords and free-text terms.

Search limits were set to English language articles, published between January 2016 and November 2022. Date limits were chosen for pragmatism and to identify the most recent evidence. Both peer-reviewed and grey source literature were included. Databases consulted included MEDLINE, Scopus, Global Health, Embase, Emcare, and Google Scholar, facilitating optimal and comprehensive coverage. Keyword searches were also performed on individual organisation websites.

Article titles and abstracts (or summaries) resulting from the initial search were reviewed and categorised according by relevance, to produce a short-list. Articles and reports not relating to human public health (e.g. veterinary or clinical focussed), to GHS/IHR or the wider context of health system strengthening for epidemic preparedness/control, or LMICs, were excluded, along with publications for which the full texts were not available. Articles that, for instance, did not refer specifically to international capability building, technical assistance or partnership working were retained and categorised as semi-relevant. Additional backward and forward citation searches identified further applicable published evidence, including pre-2016 publications. Full texts of the short-listed articles were then reviewed in detail to inductively identify key themes concerning approaches to effective IHR capacity building. A snowballing approach was used to identify further relevant publications, relating to emerging themes.

The search strategy and Preferred Reporting Items for Systematic reviews and Meta-Analyses extension for Scoping Reviews (PRISMA-ScR) checklist [26, 27] and flow diagram [28] are included in the supporting information (S1 Appendix).

## The 'What' of IHR capacity building: key factors for effectiveness and roles of international support programmes

### Public health workforce technical training and mentorship

A public health system capable of preventing, detecting, and responding to public health threats is contingent on a skilled public health workforce "from frontline to senior management" [20], adequately trained in core technical competencies [7, 29]. Insufficient workforce technical capability and training in both public health competencies and the processes and systems to support practice, are frequently cited barriers to IHR compliance and effective disease surveillance and response, factoring in the exacerbation of public health emergencies (such as COVID-19) [14, 30, 31]. Consequently, IHR-related interventions and support programmes often prioritise "technical assistance", that is, training, mentorship, or technical skills development [9, 13, 32–36].

The importance of IHR technical skills strengthening efforts has been repeatedly noted [13, 37, 38]; an "effectively trained and deployed health workforce is positively associated with addressing many health system challenges" [37]. Technical training and mentorship interventions in laboratory diagnostics, disease surveillance, and other core IHR public health capacities have actively supported improvements in workforce capability, preparedness and IHR implementation [20]. Training through Field epidemiology training programmes (FETP) have enhanced countries' capacity for disease surveillance, reporting, and outbreak response,

including through major public health crises [30, 39–42]. In Uganda, technical skills development and public health fellowship training programmes improved workforce competency and retention, contributing to system resilience and preparedness [43, 44]. In Guinea, staff training supported effective implementation of a digital health information system (DHIS2) [30], and throughout Africa and the Caribbean, laboratory personnel mentorship and training facilitated laboratory functionality and surveillance [20, 45]. Finally, health workforce training in Papua New Guinea, supported by an international organisation, contributed to cholera outbreak management [46].

The content, format, and delivery style of training impact their effectiveness. Training must target identified priority competency gaps, be directly relevant to participants' work, and appropriately tailored to the context and culture–thus co-development is vital [9, 20, 41, 47]. Consideration of delivery approach is important; for instance, whilst online training and learning platforms can be particularly beneficial in resource-limited settings, face-to-face interactive training may be more culturally appropriate and permissive to local contextualisation [37, 47]. A review of capacity building approaches in North Africa and the Middle East highlighted the importance of hands-on practical training in addition to theory [37]. "Learning by doing" has been emphasised in various contexts, including a short-term exchange programme supporting staff from both South African and UK national public health institutes to develop their epidemiology and laboratory skills [41, 48].

International support programmes play a recognised role in sharing and developing technical expertise for IHR core competencies. Sustainable skills development is crucial, for instance through refresher training, mentorship, sub-national cascade training and "training of trainers", or local-level capacity building [30, 31, 44, 49]. However, training or "technical skills development" based interventions often are delivered in isolation from sustainable knowledge transfer or other supportive structures, failing to recognise systemic and structural factors which cause and compound IHR-related technical capacity weaknesses [13, 20, 49–51]. Such capacity building efforts can therefore have limited effectiveness and sustainability. These systemic and structural factors are discussed in further detail in subsequent sections.

## Resourcing for health security: finance, human resources and infrastructure

Sufficient and sustained resourcing, and supportive infrastructures, are paramount for effective IHR technical capacity building [6, 20, 52, 53]. Whilst preparedness investments are arguably cost-effective [9, 44, 54], resourcing IHR capacity strengthening remains challenging for most countries [13]. Inadequate and unsustainable finance, human resources, or other resources are frequently cited barriers to public health emergency responses and sustainable IHR capacity building [13, 30, 32, 38, 45, 55–57]; resource shortages and financial underinvestment have again been highlighted through the COVID-19 pandemic [15, 58]. Increased predictable and sustainable national and international financing for IHR implementation is urgently required [14]–specifically including financial investment in public health infrastructure and IHR core capacity development [6, 13, 31, 53, 59].

Nationally, this requires long-term financial planning and dedicated budgets, developed collaboratively between government and national stakeholders across sectors [13, 20, 31, 54, 60]. Several authors have highlighted the important role of external or international financing, and the shared responsibility of state parties (including HICs in line with IHR commitments), international development partners and WHO to facilitate and sustainably finance IHR core capacity strengthening [6, 13, 14, 20, 25, 61]. Shaphar et al. also flagged the importance of

World Bank or regional development bank financial support, through specific public health programmes [9]. However, such "global solidarity" remains insufficient [62].

However, dependence on potentially in-flexible international financing is problematic and unsustainable [6, 12, 20, 38, 43]; the international "development assistance" model is inadequate for financing preparedness [54]. Donor-driven funding may "crowd out" and undermine moves towards sustainable domestic resourcing [63], disproportionally influence national priority-setting [63], and divert workforce and resources from essential health system functions [6, 12, 38, 52]. An influx of external funding during public health emergencies, whilst sometimes paving the way for public health system capacity development [30], can often take a short-term view, prioritising temporary infrastructure development and narrow-focussed staff training at the expense of longer-term strengthening [57]. Vertical investments often prioritise limited sections of the health system or specific diseases, generating silos and parallel systems, and limiting the potential for building GHS capacity [12, 36, 63]. Reliance on donor funding also jeopardises sustainability if funding is cut [12, 30]. Internationally supported capacity building interventions themselves can also be resource intensive and difficult to sustain long-term [20, 49, 51].

There is concern on the tendency for some donors to focus on more tangible or "visible" support over strengthening supportive infrastructures, including administrative, management, and operational systems–which are crucial for sustainable health gains [9, 63]. Dieleman et al. presented the case for meaningful partner investment in health system enablers and 'pillars', including public health workforce development, and systems and networks for laboratories, surveillance, health information, financing and management [63]. Establishment of an effective laboratory coordination network and digital information management system were pivotal to rapid expansion of COVID-19 testing capacity in Ethiopia, in addition to laboratory staff technical skills-building [64]. Khan et al. further described the importance of robust information systems for epidemiological data usage and sharing, supporting timely local responses [4]. Similarly, Reynolds et al. highlighted the importance of digital health information systems for timely reporting, as well as the challenges implementing such systems in Guinea due to physical resource constraints (e.g. internet connectivity) [30].

For successful IHR implementation countries must prioritise public health infrastructure and human resource development [3, 57]. Lack of both adequate infrastructure and trained personnel—stemming from financial constraints—have been contributors to rapid escalation of public health emergencies such as the 2014–2016 West Africa Ebola epidemic [13]. Health workforce-related constraints were also experienced in many countries through the COVID-19 pandemic–for instance insufficient staff number and uneven geographical distribution [15]. Kandel et al. similarly noted the dependence of effective outbreak response on the availability of human resources, financing, and effective logistics management [5].

Limited professional job opportunities for trained staff (e.g. FETP graduates) within the national system [41], low staff retention and rapid turnover frequently cause human resource shortages, undermining sustainable workforce capacity development in LMICs [20, 49, 51]. Loss of skilled public health professionals from the national system may be exacerbated by better employment opportunities and salaries offered in HICs or HIC-based agencies.

Human resource limitations also hinder the effectiveness of public health and IHR capacity development programmes in LMICs [30, 36]. A study investigating the impact of disease surveillance training on reporting found that its effectiveness was limited by broader budgetary and infrastructural constraints, competing staff work-loads, and health worker strikes [51]. Workforce strengthening investments therefore need to go beyond stand-alone technical training [15], recognising and where possible addressing the sources of human resource shortages. For instance, training that is continuous, inclusive and integrated within career

development pathways can promoted [57, 65]. In Lebanon, policies investing in health worker training, career path development and improved salaries contributed to increasing workforce numbers [57].

External donors and international support programmes play an important role in directly resourcing public health capacity building [25, 44], sharing expertise and best practice on sustainable financing [38, 52], mobilising political support [20], and leveraging domestic funding [63]. However, there is recognition of the need for health-related international support to adopt revised models of financial support, working through and supporting development of national systems, infrastructures, and human resources [25, 63]. Investments should be aligned to local priority needs, collaborative and flexible to integrate within the wider system and reduce inefficiencies, and advocate and plan for sustainable domestic resourcing [9, 12, 30, 31, 63]. The integration of IHR implementation strengthening investments within national system and budgets, as promoted through national action plans for health security (NAPHS), is crucial [12, 53]. Montgomery et al. similarly highlighted the importance of "building for flexibly" within GHS capacity development efforts, through investing in core public health capacities and systems capable of responding to all hazards [36].

## High-level political support and leadership

Kluge et al. stated that of the six WHO health system building blocks, "Leadership and governance" is likely "the most important in improving IHR implementation" [53]. Sustained political commitment and leadership at regional, national, and institutional levels have consistently facilitated multisectoral coordination [55, 60, 66], public health system development, strengthened IHR implementation [13, 20, 53, 59, 67, 68], and effective outbreak management [38, 46]. For instance, commitment to IHR strengthening from African heads of state and the launch of Africa Centres for Disease Control and Prevention (Africa CDC) in 2017, developed momentum and steer for continent-wide IHR strengthening [59, 69]. Reynolds et al. also remarked on the requirement for sustained political will across different levels of government for DHIS2 rollout in Guinea, due to the size and transformational nature of the work [30]. The WHO Committee report on the functioning of the IHR during COVID-19 likewise asserted that IHR implementation responsibility sits at highest levels of government [14].

High-level leadership and coordination, and financing and resourcing for IHR implementation, form two of the 19 IHR technical areas [2, 7, 29]–and the two are arguably intrinsically linked [13]. Suthar et al. note that "health security is a continuous process in which action, financing, partnerships and political commitment must be sustained" [20, 60]. In calling for the strengthening of health systems and health security in Africa, Nkengasong et al. emphasised that political commitment must be established and translated into the release of domestic financing [59].Whilst large-scale public health emergencies stimulate high-level support and financing [20, 38], lack of political leadership and financing together consistently emerge as barriers to longer-term IHR implementation and health system strengthening [13, 14, 70]. Even during the COVID-19 response, political support and resources for IHR implementation remained "insufficient and irregular" at national and international levels [11]. Commenting on the utility of the current IHR in the wake of COVID-19, Hannon et al. described "the underlying problem of a lack of broader political will, including to commit resources that could improve core capacities" [71].IHR National Focal Points (NFPs) have a lead role in IHR coordination and implementation, and emergency response [2, 13, 29, 72]; empowerment of such IHR leadership functions is crucial [11, 13]. Experience from the H1N1 pandemic, the West Africa Ebola epidemic, and COVID-19 shows that NFPs should be positioned at a senior level, with access to multiple sectors and robust resourcing, technical, and political capabilities

[11, 13]. However, insufficient authority, leadership, training and resources have limited NFP capacity to fulfil their functions under the IHR [13, 29, 55, 72], undermining effective IHR implementation [11].International partners and programmes can support in advocating for high-level national commitment and leadership for IHR implementation, establishing IHR "champions", and the establishment, operationalisation and capacity building of public health agencies and NFPs [7, 29, 38, 55, 63]. Promoting country ownership and leadership for specific IHR core capacity implementation is also important to ensure sustained development [31]. The importance of leadership and management skills development, and "softer" public health competencies, is also increasingly recognised, but often overlooked in technical skills-focused capacity building efforts [9, 25, 50, 73]. International partners can invest in leadership development and governance strengthening at individual, institutional, community, or national levels. Leadership and management training, through FETPs, laboratory leadership programmes, and hazard management programmes, have equipped professionals in several countries to manage public health events, contributing to improved disease preparedness and response, and IHR compliance [20, 41, 59].

## Legislation, frameworks, strategies and plans

The IHR (2005) provide the mandate and legal framework for GHS [2, 6]. COVID-19 and other major public health events have necessarily prompted assessment of their continued suitability and sufficiency [11, 52, 74]. Whilst the IHR remain a "cornerstone of international public health [11]", promoting timely detection and response to complex public health emergencies [20, 38, 74, 75], they have recognised limitations which have been brought into the spotlight through the COVID-19 pandemic [6, 14, 23, 76]. A review of the IHR during the COVID-19 response found that more effective implementation is sorely needed–with significant gaps in preparedness particularly highlighted [11, 14, 62]. Bartolini also noted a need for clearer definition and emphasis on international financial and technical support obligations for core capacity development, within the IHR [62]. These points illustrate the inability of the IHR to hold states to account in terms of compliance or transparency of core capacity reporting and information exchange, or to ensure global coordination and governance [23, 71, 77]. Assefa et al. argued that factors such as health system fragmentation and socio-economic inequity strongly influenced the differential impacts of COVID-19 between countries, in terms of the case/death rates [24]. However, the current IHR are "state-centric", failing to account for societal inequities within countries as well as global inequities, "assuming parity in health systems across the globe". Thus Šehović and Govender recently called for an equity focus within the IHR, for instance a framework to highlight inequities and promote effective pandemic control [16]. At this point in time the discussions and proposals for amendments to the IHR continue [23]. The shortcomings in the IHR have also prompted calls for a separate United Nations-level pandemic treaty [71, 77].

Several international frameworks and strategies for GHS and IHR implementation exist [7, 29, 66, 78], which establish standards and guidance for system-level capacity development across the IHR technical domains [5, 9, 20]. Practical tools such as JEEs, IHR State Party Self-Assessment Annual Reports (SPAR), the WHO Benchmarking tool, simulation exercises, and risk assessments frameworks and targets, enable the evaluation of capabilities and preparedness, support prioritisation of urgent actions, provide standardised "goals and metrics to track progress" [36], and foster multisectoral collaboration for improved IHR implementation [8, 9, 29, 36, 38, 79, 80]. Recent epidemics have highlighted the importance of such instruments in health systems strengthening [13, 53], although questions have arisen through the COVID-19

pandemic on the reliability of the SPAR and other indicators are as predictors of preparedness, considering how some higher-scoring countries have fared [81, 82].

At a national level, legislative and regulatory frameworks provide fundamental backing for sustainable public health capacity development and may establish practices which outlast government or regime changes [83, 84]. They can also empower IHR NFPs to fulfil their mandate [13, 14].

Following the West Africa Ebola outbreak response, the need for financed multi-level strategic plans to improve IHR implementation was emphasised [13]. Preparedness gaps highlighted by the escalating COVID-19 outbreak in early 2020 caused Nkengasong and Mankoula to advocate for "a unified [and comprehensive] continent-wide strategy for preparedness and response" and committed financing "to support country-customised implementation plans" in Africa [85]. In examining the attributes of health systems crucial for resilience to shocks, Grimm et al. also noted the importance of both formal channels for integrated coordination and national strategies–implemented at all levels—for effective emergency preparedness and response [57]. Orelle et a. further describe elements essential to developing biosafety and biosecurity capacity, including the development and/or adaptation of national policies, guidelines, regulations and standard operating procedures [31].

Strategic plans thus help to set direction and promote standardised, validated practices, and sustainable capability building [13, 20, 86].However, insufficient implementation of such strategic plans has repeatedly been noted as a barrier to building IHR capacity [31, 57]. In the context of effective responses to COVID-19, the importance of synergy and integration between IHR-related national plans and other national health plans–particularly around universal health coverage (UHC) and primary care, the contextualisation of national plans and strong governance, were also highlighted [24].

National pandemic preparedness and disease-specific plans should also align with international plans and frameworks for GHS "as part of a single platform" [11, 12]. Human resource mapping and planning, including strategies for workforce recruitment, development, (re-) training and retention, are equally key for sustainable capacity building, workforce resilience, and emergency response [20, 46].

To promote national ownership, effectiveness, and long-term sustainability, international support should align with (inter)national frameworks, strategies, and operational plans for IHR capacity development. Informed by needs assessment, it can also partner in the development and implementation of contextualised strategic plans and frameworks [87], including management, human resources, and administration system planning [9].

## IHR compliance, GHS and health system strengthening

IHR core capacities align closely with essential health system-wide functions and "building blocks" [53] and effective public health emergency response is dependent on functional and resilient health systems [15, 20, 57]; health system weaknesses, paralleling IHR capacity weaknesses, have been argued to impede IHR implementation [59] and exacerbate the spread and severity of epidemics [14, 15]–demonstrated most recently with the COVID-19 pandemic [24].

Thus, developing IHR capacity and pandemic preparedness, health systems strengthening and achieving universal health coverage (UHC) are intrinsically linked and interdependent [24, 53, 63, 88]. The COVID-19 pandemic has brought renewed insights into these interdependencies, and fresh calls for prioritisation of UHC to strengthen both health systems and GHS [16, 89]. In their analysis of countries' responses to COVID-19, Assefa et al. observed that "Countries with health systems and policies that are able to integrate International Health Regulations (IHR) core capacities with primary health care (PHC) services have been effective at

mitigating the effects of COVID-19" [24]. However, Lal et al. [17] argue there has been under-appreciation of the importance of UHC and primary care for public health emergency preparedness and response; states have prioritised GHS to the detriment of attaining UHC, although "effective and accessible primary health care can be a key approach for creating cohesion between global health security and UHC". Underinvestment and neglect of UHC (and provision of high quality, accessible and acceptable health services) have resulted in weak, fragmented and inequitable health systems. The COVID-19 pandemic exposed these weaknesses, with inadequacies in primary care undermining the response. Lal et al. thus propose means for aligning and integrating GHS and UHC efforts to strengthen health systems. The recent Singapore Statement on GHS similarly declared: "GHS is made sustainable only when embedded within universal health systems" [65]. In a recent Lancet Comment, Kishida advocated for human-centred "human security" approaches and prioritisation of UHC to equip health systems to respond to health challenges–combining efforts to enhance preparedness with efforts to strengthen primary care [90]. The 2021 IHR Review Committee also emphasised integration of emergency preparedness, surveillance, and response core capacities "within the broader health system" to ensure resilience of health system function during health emergencies [14].

Health system resilience has been described in terms of the ability to withstand, absorb, adapt and respond to shocks, maintain core functions, and both learn and transform services through crises [91]. Viewing health system resilience through the WHO's Six Building Blocks of a Health System framework [92], may offer a useful perspective by which health system leaders and managers at all levels, as well as international partners, can systematically assess health system resilience and GHS, and map out priorities for action. Through this framework's lens, a resilient health system is also one which has capacity to ensure and sustain access and coverage of essential services, and quality and safety of care. Echoing the WHO African Region Framework for Health Systems Development, Karamagi et al. similarly characterise health system functionality–working towards UHC–in terms of four facets: "access to, quality of, demand for essential services" and health system resilience [93]. Along a similar vein, Nkengasong et al. proposed five key health system improvements for health security in Africa echoing the WHO Building Blocks–including strengthening and enabling public health capabilities, workforce, and governance [59].

An Organisation for Economic Co-operation and Development report highlighted the importance of international support which "protects and reinforces existing health priorities" [38]; interventions to strengthen IHR implementation and health security have supported health system strengthening and vice versa–reflected in improved health outcomes [13, 20, 53]. However, international capacity-building interventions frequently employ vertical, disease-specific approaches. During public health emergencies, donors and NGOs can also tend to focus on immediate assistance at the expense of health system strengthening [57]. This can cause accumulation of parallel and temporary systems running independently to the health system–undermining sustainable health system strengthening [25, 63] as well as impeding outbreak response [30]. In view of the impact of COVID-19 and unsustainability of vertical approaches, El Bcheraoui et al. proposed prioritisation of "a new model of development assistance for health. . . focused on stronger and more resilient health systems" [25]. Kluge et al. noted the importance of joint-working for health system strengthening and IHR implementation [13, 20, 53], also citing the IHR MEF assessment of both health security capacity and broader health system functioning [7, 53, 66].

Thus, international IHR-related interventions should be embedded within and help develop the capacity and resilience of the existing health system, promoting attainment of UHC [13, 15, 53, 94]. The WHO acknowledged the need for international partner-led health system

strengthening interventions to prioritise IHR capacity strengthening–ensuring core capacity development goes "hand-in-hand with overall strengthening of the health system" [13].

Notably, resilient health systems and effective international public health surveillance and response require both national and sub-national capacity and coordination [65, 95]. However, it has frequently proved challenging to make substantial progress sub-nationally for improved system-wide IHR implementation [3, 20, 30]. Dedicated support to build sub-national capacity and develop functional linkages between sub-national and national levels are also important for sustainable IHR and health system strengthening [3, 20, 95]. Building international collaboration and networks can also support IHR capacity strengthening and GHS, and support programmes can again play a facilitative role here [36, 57].

## The 'How' of IHR capacity building: Partnerships and technical collaborations

### Partnership principles

Fostering "global partnerships" is crucial for successful IHR implementation and GHS [2, 36]. Following the West African 2014–2016 Ebola epidemic, a WHO report cited the importance of partnerships "with communities, between countries, within regions, with development and aid organizations, and with WHO" for IHR implementation, and to ultimately improve preparedness and response globally [13]. Evidence has demonstrated the importance of not only *what* public health and IHR capacity building support delivers, but *how* it is delivered. The characteristics and working principles of these partnerships and programmes—including bilateral and multilateral partnerships, WHO assistance, or support from international aid agencies, NGOs, civil society, or academia–are paramount [96].

Effective partnership-working has facilitated public health capacity development for improved surveillance and outbreak response. In response to Zika virus spread in 2016, the Government of Vietnam increased surveillance and response capacity through technical partnership with organisations including WHO and the US Centers for Disease Control and Prevention (US CDC). Creation of a national Emergency Operations Centre facilitated data analysis and sharing amongst partners nationally and internationally [97]. Across South East Asia, inter-country collaboration and technical partnerships, with development of strong formal and informal trusting relationships, facilitated the development of operational preparedness and response guidelines, and the sharing of information via regional platforms [98]. Similarly, Montgomery et al. cited the importance of building trusted partnerships, through long-term engagement, as a key lesson facilitating success in a GHS capacity building programme and subsequent outbreak response coordination [36]. Respectful partnerships have also been emphasised as a key pillar of the Africa CDC-defined New Public Health Order for Africa [99].

In the context of partnership-working during the 2014–2016 Ebola epidemic response, Standley et al. emphasised similar themes, namely the importance of collaboration, careful stakeholder involvement, and consensus decision-making, as well as lasting relationships and trust to ensure the sustainability of investments beyond the immediate emergency support period [67].

Effective bilateral partnerships have been recognised to establish collaborative approaches [67], facilitating regular communication, transparency, trust, and the multidirectional transfer of knowledge and learning between partners; this translates into long-term mutually beneficial relationships with equitable inputs and opportunities for both HIC and LMIC partners [36, 48, 67, 100]. Wilson and Cartwright attested to the value of trusted peer-to-peer partnerships and a "doing with" rather than "for" approaches to capacity development, in a bilateral partnership

between the UK IHR Strengthening Project and the Pakistan government. This led to a fusion of experiences and technical expertise, and "co-production of sustainable solutions" [50].

However, global health partnerships and programmes can commonly be compromised by factors such as limited local engagement and cultural insensitivity from international/HIC partners, or adoption of "one-size-fits-all" approaches inflexible to local contexts and health systems. Involvement of a multiplicity of external partners—combined with poor coordination —can also be potentially duplicative and resource-intensive for national partners [44, 67, 101].

Khan et al. [102] and Plamondon et al. [96] outlined how HIC-based global health organisations can perpetuate inequities rather than promoting effective partnerships, for instance: maintaining unequal distribution of staff and resources (e.g. solely HIC-based); embarking on interventions or research with little coordination or engagement with recipient communities; and limiting participation of LMIC expertise in governance whilst using HIC-sourced "experts" with limited project leadership experience. The legacy of colonialism especially detrimentally affects international support and partnership-working. International programmes and their priorities often reflect historical ties and regions of strategic or geopolitical importance for HICs, rather than LMIC priorities. Bilateral programmes between organisations from former colonial powers and ex-colonised nations can exhibit unbalanced power dynamics, impeding equitable partnerships [96, 103].

COVID-19 has again spotlighted these ongoing inequitable power differentials between the global "north" and "south", along with nationalism and extractive partnerships instead of global solidarity expressed by HICs [77, 104]. This has been illustrated explicitly in the inequity of vaccine availability between HICs and LMICs—despite vaccine trials having often been conducted in LMICs [16, 24, 105]. These inequities persist despite recognition that such inequities undermine GHS and pandemic recovery [16, 24].

The impacts of colonialism and its "complex interdependence. . .with health, economic development, governance and human rights" [106] must be openly acknowledged and examined, along with inequitable power and political economies, and the "equity implications of all aspects of partnering" [96]. In the context of bilateral partnerships, "powers-turned-partners" play a pivotal role in efforts to decolonise global health, prioritising and working towards equity-centred, inclusive, and effective partnerships for global health; such relational partnerships would emphasise "social justice, humility and mutual benefits" [96].

More broadly, COVID-19 has shown the requirement for changes in global governance to decolonise and address global inequities, including for instance an equity focus within the IHR [16, 24, 77, 104]. Büyüm et al. also note that whilst the COVID-19 pandemic has spotlighted and reinforced injustices, an opportunity is presented to transform global health through re-politicising and re-historicising global health, elevating "global south" leaders whilst addressing gender disparities, and promoting bidirectional knowledge flow—with contributions from the "global south" driving policy-making and practice [106]. Such fundamental shifts can also be integrated within international and bilateral partnerships for IHR strengthening. For instance, "twinning" between respective Ministries of Health or national public health institutes has been recommended "as part of an integrated approach to health systems strengthening" [13].

## Multi-sectoral partnerships and One Health

Non-health sectors are impacted by and crucial in public health emergency preparedness and response [11, 107]. With an estimated 75% of emerging infectious diseases being of zoonotic origin [108], effective multisectoral communication and coordination at all levels, particularly

involving animal health, are essential to GHS and health system resilience [15, 24, 65]—characterised by adaptiveness and flexibility to respond to diverse public health threats [36, 57].

The IHR (2005), JEE, World Organisation for Animal Health Performance of Veterinary Services (PVS) pathway and other frameworks apply multisectoral coordination approaches and provide tools to assess collaboration across animal and human health [2, 7, 29, 55]. The IHR-PVS national bridging workshop forms one such mechanism designed to bring human and animal health sectors together to jointly assess gaps and plan for collaborative action for improved human and animal health security [109]. However, JEE assessments in many countries have revealed considerable gaps between human and animal health sectors, highlighting the need for "One Health" approaches which, recognising the interrelationships between animal health, human health and the environment, seek to improve "coordination, collaboration and communication at the human-animal-environment interface to address shared health threat" [10, 88, 109].

National Action Plans for Health Security (NAPHS) have also helped facilitate improved multisectoral coordination and One Health perspectives [5, 94]; in Tanzania, a collaborative multisectoral One Health approach was taken to NAPHS development. As well as helping to identify technical and funding gaps, the process generated enabling partnerships across sectors, and with other countries and international partners [15, 66].

International partnerships have also played a role in facilitating cross-sector engagement for improved preparedness. In Ethiopia, multisectoral One Health collaboration between the Ethiopia Public Health Institute and other government departments, with support from international partners, facilitated development of the national Rabies control plan and two mass Rabies vaccination campaigns [110]. The US Global Disease Detection capacity building programme similarly adopted One Health approaches, working to promote multisectoral collaboration to strengthen "all hazards" response capacity across the countries it engaged with [36].

Through the COVID-19 pandemic the importance of prioritising multisectoral One Health approaches for preparedness and response, has once more been emphasised [11]; in response to the pandemic, Africa CDC likewise advocated for One Health approaches and the sharing of best practice, improving multisectoral collaboration for increased disease surveillance [107]. However, globally, robust intersectoral coordination and collaboration and zoonotic disease preparedness remain inadequate [5, 80].

The 'how' of international IHR capacity building efforts therefore goes beyond effective bilateral partnership working between health sector institutions [55]. Partnerships should exhibit approaches which are "practical, sustainable, collaborative, and based on the [country] needs", prioritising multisectoral collaboration and One Health approaches, with equitable partner coordination at international, national, and sub-national levels, to promote sustainable development [3].

## Conclusion

This review has examined evidence from a range of LMICs and international support programmes–looking at both successes and challenges of capacity building efforts, and experiences from public health emergency preparedness and response efforts. An exploration of emerging themes has highlighted several crucial factors to promoting effective and sustainable improvements to IHR compliance–which ultimately depend on the development of strong public health systems. These facets of sustainable IHR and whole health system strengthening are conceptualised schematically in Fig 1.

The themes discussed provide important lessons for international partners and programmes, who can play an important role in supporting improved IHR implementation, yet whose perspectives and approaches can enhance or undermine success. Several recommendations have

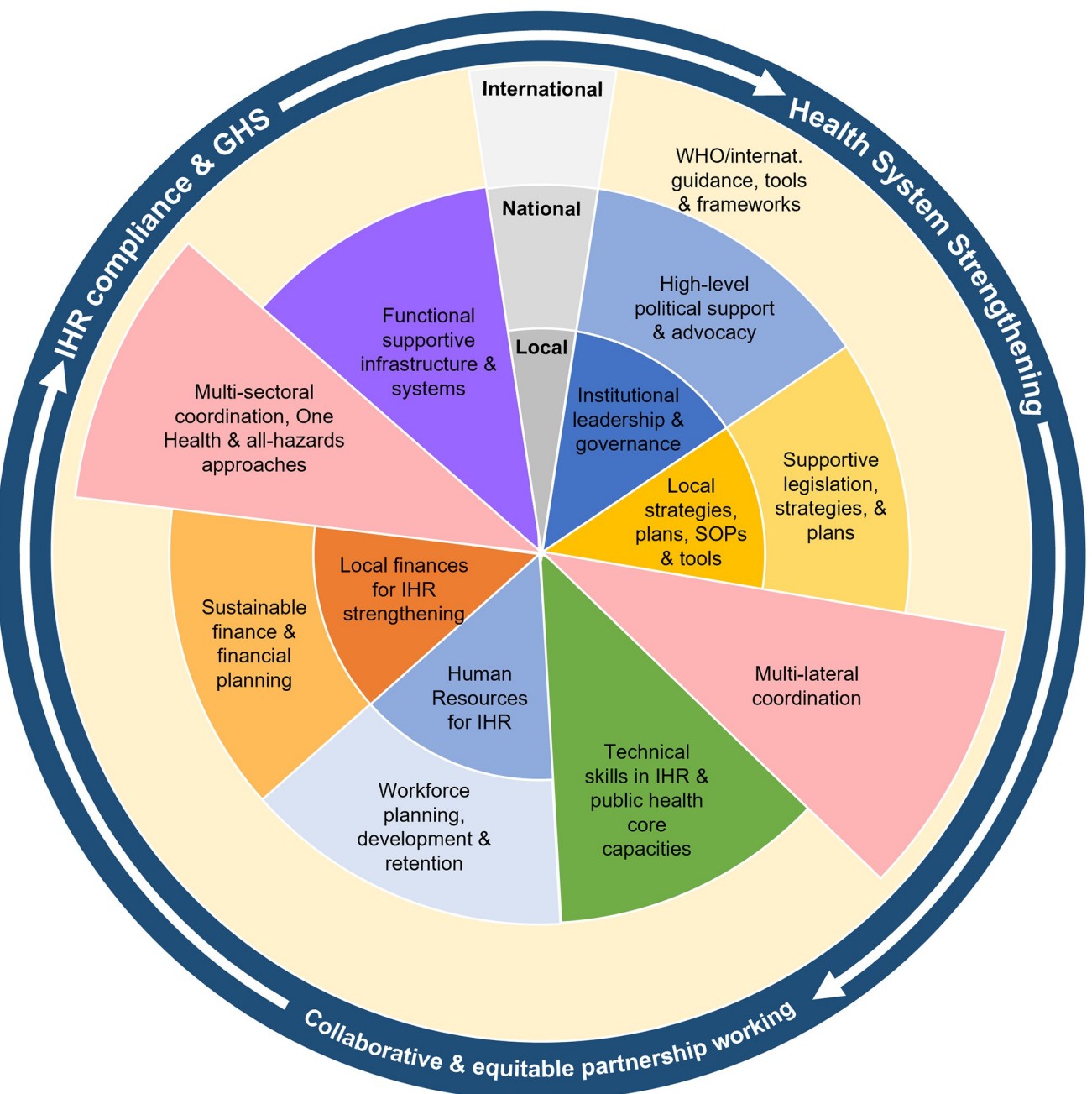

**Fig 1. Conceptualisation of important factors and approaches to global health security capacity building.** Schematic representation of emerging themes: key elements for sustainable GHS and IHR capacity building and health system strengthening operating at local, national and international levels.

therefore been drawn for GHS support programmes, as described in Table 1. Some lessons will also likely be applicable to international support initiatives more generally.

Individual IHR capacity building efforts and international support programmes are unlikely to be able to support all of the factors illustrated in Fig 1, or to work across all levels of the system. However, recognition and consideration of these factors–and the interplay and interdependencies between them–is important and should inform engagement. Strong partnership-centred working is paramount, and the *how* of international support is arguably

**Table 1. IHR capacity building perspectives and approaches: Recommendations for international support programmes.**

| Key factors for sustainable GHS capacity building | Recommendations for international partners |
| --- | --- |
| IHR and public health technical skills development (e.g. training) <br> Public health workforce, leadership and management development | • Technical skills development should move beyond a focus on narrow cadres of the public health workforce (e.g. field epidemiology training programmes), to cover a breadth of public health personnel and capacities, for instance guided by international public health competency frameworks. Mechanisms for the dissemination of training and maintenance of technical skills should also be considered. <br> • Training interventions should be co-developed with partners, culturally and contextually appropriate, and include relevant hands-on practice. <br> • Partners should support the development of effective leadership and management (both people and systems), e.g. through leadership and management training and mentorship. |
| Sustainable financing and financial planning <br> Investment in human resources and supportive infrastructures, systems and networks <br> Strategies, policies and plans for IHR and GHS | • Commitment to greater predictable and sustainable funding for IHR implementation is urgently needed. Whilst this requires HICs to meet IHR commitments to share responsibility for IHR compliance across state parties, it must be balanced with ensuring sustainable domestic resourcing, and recognising the limitations of short-term external funding. International support programmes may be well-placed to advocate for or share learning concerning building sustainable domestic resourcing. <br> • Greater investments should be made in the more 'invisible' but fundamental aspects of public health infrastructure and systems—supporting the development, for instance, of: strategies, networks and policies for emergency preparedness; processes for improved information flow; or career progression pathways and workforce planning to improve staff retention. <br> • HICs and international support programmes must recognise their role in the loss of skilled public health and healthcare professionals from LMICs, causing low staff retention and rapid turnover–which limits sustainable health system development. Careful planning must be undertaken by international support programmes to ensure their recruitment strategies do not unintentionally drain skills and experience from the national system. |
| Integration of IHR capacity building within health system strengthening and universal health coverage efforts <br> Alignment with local need, national strategies and priorities, and international frameworks and guidance | • IHR capacity building efforts should fundamentally be embedded within and work to strengthen the wider health system. This requires efforts to develop proper understanding of the national system, to work collaboratively with national stakeholders, and to integrate initiatives and resources with wider health system strengthening and universal health coverage efforts. <br> • Frameworks such as the WHO Six Building Blocks of a Health System can be used as practical tools to help assess need and prioritise efforts to build health system resilience. <br> • International support programmes should work to ensure their workplans and approaches–including monitoring and reporting mechanisms–are co-developed with national partners and aligned with national priorities, policies and frameworks, as well as informed by a local understanding of need and facilitating local ownership. |
| Multi-sectoral and multi-lateral coordination, One Health and "all-hazards" approaches | • Implementation of One Health approaches should be prioritised, involving intentional and robust multi-sectoral and multi-lateral engagement and coordination, and supporting mechanisms for improved data flow. This also requires flexibility, implementation of multi-faceted approaches and promotion of systems thinking. |

(*Continued*)

**Table 1.** (Continued)

| Key factors for sustainable GHS capacity building | Recommendations for international partners |
|---|---|
| Effective and equitable partnership working and addressing the legacies of colonialism | • Cross-partner coordination mechanisms are needed to ensure that trusted partnerships can be built, reducing duplication of efforts across partners and reducing the resources national partners need to invest in managing partnerships.<br>• The ongoing impacts of colonialism and history on power imbalances within global health initiatives must be openly acknowledged and examined in the international community and within individual programmes. International partnerships may require substantial reorientation to establish equitable partnerships, which embrace local expertise and priority-setting, and multi-directional learning. |

equally important to *what* it delivers. Collaborative and complementary working, multisectoral coordination, and system-wide (sub-national to international) engagement are vital. Crucially, efforts should be built upon a foundation of strong, equitable and mutually beneficial partnerships. For international support programmes, the adoption of new perspectives of partnership-working will help reverse the perpetuation of inequalities and power imbalances which have marked much international support to date. As such, efforts to decolonise global health are particularly important.

Several limitations of the review are evident. Firstly, this is not a systematic review and relevant articles may have been missed, for instance those not referring specifically to the IHR or GHS. The evidence strength and quality were also not assessed in depth; the focus was on identifying emerging themes from a range of sources and literature, considering the effectiveness and sustainability of international support, as well as narrative observations and insights from such work. A further limitation was the exclusion of non-English language studies and possible underrepresentation of LMIC-authored studies or articles–which itself illustrates the inequalities of predominantly HIC-directed perspectives. The primary authors of this article are UK-based, with subsequent work intended to focus on LMIC perspectives.

This review focused on LMIC settings, although as the COVID-19 pandemic has demonstrated, both HIC and LMICs are underprepared to deal with public health emergencies [9, 24]. Effective GHS evidently requires more than strong "technical" IHR capacity. This emphasises the need for equitable system-wide approaches, and further research and evidence generation especially from LMIC perspectives. Haldane et al. pointed to the need for increased receptiveness to knowledge and expertise from LMICs; HICs have much to learn from LMICs to achieve health system resilience, and effective preparedness and response [15].

## Supporting information

**S1 Appendix. Literature review search strategy and results, PRISMA-ScR checklist and flow diagram.**
(DOCX)

## Acknowledgments

We would like to acknowledge UKHSA Knowledge and Library Services for their support in refining the search strategy and conducting the searches, and Sarah Shanks for initial support in gathering evidence.

## Author Contributions

**Conceptualization:** Anne Doble, Ebere Okereke.

**Data curation:** Anne Doble, Zoe Sheridan.

**Formal analysis:** Anne Doble, Zoe Sheridan.

**Methodology:** Anne Doble, Zoe Sheridan.

**Supervision:** Ahmed Razavi, Anne Wilson, Ebere Okereke.

**Visualization:** Anne Doble.

**Writing – original draft:** Anne Doble, Zoe Sheridan.

**Writing – review & editing:** Anne Doble, Zoe Sheridan, Ahmed Razavi, Anne Wilson, Ebere Okereke.

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
