## [Decision Letter · Decision Letter 0]

24 Jan 2023

PGPH-D-22-01694

The role of international support programmes in global health security capacity building: an evidence review

Dear Dr. Doble,

Thank you for submitting your manuscript to PLOS Global Public Health. After careful consideration, we feel that it has merit but does not fully meet PLOS Global Public Health’s publication criteria as it currently stands. Therefore, we invite you to submit a revised version of the manuscript that addresses the points raised during the review process.

The manuscript represents a timely and important piece. Please ensure that the topic of animal health is acknowledged in the revised submission, and that terms are defined clearly in the early sections; this will add to the conceptual framing. The reviewers have also made additional comments which will likely increase the clarity and depth of the paper, so I encourage you to consider them closely. 

We look forward to receiving your revised manuscript.

Kind regards,

Claire J Standley

Academic Editor

Journal Requirements:

1. Please identify the study as a scoping review in the title.

2. Please send a completed 'Competing Interests' statement, including any COIs declared by your co-authors. If you have no competing interests to declare, please state "The authors have declared that no competing interests exist". Otherwise please declare all competing interests beginning with the statement "I have read the journal's policy and the authors of this manuscript have the following competing interests:"

3. Please indicate in the online submission the full and correct funding information for your study and confirm the order in which funding contributions should appear. 

4. We do not publish any copyright or trademark symbols that usually accompany proprietary names, eg (R), (C), or TM  (e.g. next to drug or reagent names). Please remove all instances of trademark/copyright symbols throughout the text, including R on page 5.

5. We have noticed that you have uploaded Supporting Information files, but you have not included a list of legends. Please add a full list of legends for your Supporting Information files after the references list. 

Additional Editor Comments (if provided):

The manuscript presents a timely and important analysis of health security capacity strengthening and the role of international support programmes. The peer reviewers have made some pertinent suggestions for further strengthening the manuscript's arguments, as described in more detail below. I would particular recommend addressing the points raised by Reviewer #1 with respect to acknowledging animal health as a critical domain within health security, and per both reviewers, providing additional definitions in the opening sections for clarity. Please also review Reviewer #2's suggestions regarding additional references to support the description of conceptual linkages between health security, UHC, etc. The other detailed comments should also be reviewed and considered, as they will likely also provide benefit, if space allows!

Reviewers' comments:

Reviewer's Responses to Questions

**Comments to the Author**

1. Does this manuscript meet PLOS Global Public Health’s publication criteria? Is the manuscript technically sound, and do the data support the conclusions? The manuscript must describe methodologically and ethically rigorous research with conclusions that are appropriately drawn based on the data presented.

Reviewer #1: Yes

Reviewer #2: Yes

2. Has the statistical analysis been performed appropriately and rigorously?

Reviewer #1: N/A

Reviewer #2: N/A

3. Have the authors made all data underlying the findings in their manuscript fully available (please refer to the Data Availability Statement at the start of the manuscript PDF file)?

Reviewer #1: Yes

Reviewer #2: Yes

4. Is the manuscript presented in an intelligible fashion and written in standard English?

Reviewer #1: Yes

Reviewer #2: Yes

5. Review Comments to the Author

Reviewer #1: This is a well written review that adresses a topical issue in global health security. Given that most recent epidemics originate from the human-animal-enviornmental interface, I find it rather surprising that the review did not consider animal health. I also do feel that this review should make more pragmatic recommendations on the perspectives for the future. For example what needs to be done with respect to workforce development including having staffing norms by cadre? What are the orphan cadres that need to be trained? what curriculum review needs to be done? On they issue of integratation of health sucurity and resilient health systems what needs to be done? For a health manager, resilience needs to be viewed from the perspective of their core focus - delivery of essential health and related services that people need. A health manager focuses on ensuring the health investments are organized and managed in a manner that maximizes access to essential services by their recipient population. Thus, resilience needs to be contextualized within the description of a functional health system for a health manager. For most health actors, a functional health system is described from the perspective of its ability to ensure access to quality essential services that are demanded by the population. Capacities needed are therefore access to services and quality of care on the supply side, with community engagement and needs for essential services on the demand side of outputs. A description of system functionality that integrates resilience has been defined as one with the ability to ensure access to quality essential services that people are demanding for, during routine and in shock events. Resilience, together with access, quality of care and community demand represent the four distinct outputs of a functional health system. Looking from this lens of a health manager seeking a functional health system, a more practical description of health systems resilience would be the ‘ability to sustain expected access to quality essential services even when threatened by shock events’. I propose that the authors offer some insights into thios view of systems resilience.

Reviewer #2: Congratulations to the co-authors for a timely review on this topic! The paper offers a useful overview for GHS practitioners, and importantly, focuses operationalizing this work within LMICs through international collaboration. Some suggestions to help strengthen the article:

- The introduction would benefit from some important context critiquing WHY current GHS/IHR efforts have failed, in order to better understand the rationale for this piece at this specific moment

- Background sections would also benefit from defining some core concepts, as many terms in this paper have somewhat contested definitions for different types of stakeholders (eg, "public health capacity," "health system resilience," "international support programmes")

- The intro somewhat glosses over the IHR Strengthening Project, even though this appears to be a central focus of the paper. Explaining more about this as well as how it serves as an example for other types of international support programmes may offer helpful context

- The noted synergy between GHS and UHC is important, offering new ways to conceptualize effective and sustainable public health capacities, but is not explained well-enough given the relative complexity and novelty of this emerging area of research. The paper could be strengthened by more comprehensively fleshing out what researchers in this space have been arguing (eg, https://www.thelancet.com/journals/lancet/article/PIIS0140-6736(23)00014-4/fulltext, https://www.thelancet.com/journals/langlo/article/PIIS2214-109X(22)00341-2/fulltext, https://www.thelancet.com/servlet/linkout?suffix=e_1_4_1_2_78_2&dbid=4&doi=10.1016/S2214-109X(22)00341-2&key=10.1016%2FS0140-6736%2821%2900029-5&cf=fulltext&site=lancet-site, https://www.uhc2030.org/news-and-stories/news/action-on-health-systems-for-universal-health-coverage-and-health-security-555531/) -- and why it is important to helping reimagine health systems in the context of health security.

- The structure of the paper could be improved by a brief paragraph detailing the organization and conceptual framing of these topics in the main portion (eg, did you categorize them along IHR core capacities, HSS building blocks, or something else? etc.)

- While an effective review of this space, some of the potentially-most insightful information has not been fleshed out or given adequate nuance to provide a novel analysis. This could be addressed by drawing out more specific examples and unique ideas, rather than largely resorting to high-level framings. For example: "However, training or “technical skills development” based interventions often are delivered in isolation from sustainable knowledge transfer or other supportive structures, failing to recognise systemic and structural factors which cause and compound IHR-related technical capacity weaknesses [13, 17, 47-49]." -- what are the systemic and structural factors? This seems to be the most compelling finding, but is missing entirely.

- The conclusion may benefit from having more concrete recommendations (perhaps just by better organizing/structuring this section) for international practitioners to better conceptualize this work.

6. PLOS authors have the option to publish the peer review history of their article (what does this mean?). If published, this will include your full peer review and any attached files.

**Do you want your identity to be public for this peer review?** For information about this choice, including consent withdrawal, please see our Privacy Policy.

Reviewer #1: **Yes: **Dr Ambrose Otau Talisuna

Reviewer #2: No

---

## [Editor Report · Decision Letter 1]

8 Mar 2023

The role of international support programmes in global health security capacity building: a scoping review

PGPH-D-22-01694R1

Dear Dr Doble,

We are pleased to inform you that your manuscript 'The role of international support programmes in global health security capacity building: a scoping review' has been provisionally accepted for publication in PLOS Global Public Health.

Best regards,

Claire J Standley

Academic Editor

Thank you for the clear and considered responses to the previous round of comments.